# Batch-produced, GIS-informed range maps for birds based on provenanced, crowd-sourced data inform conservation assessments

Ryan M. Huang[1], Wilderson Medina[1], Thomas M. Brooks[2,3,4], Stuart H. M. Butchart[5,6], John W. Fitzpatrick[7], Claudia Hermes[5], Clinton N. Jenkins[8,9], Alison Johnston[7], Daniel J. Lebbin[10], Binbin V. Li[1,11], Natalia Ocampo-Peñuela[12], Mike Parr[10], Hannah Wheatley[5], David A. Wiedenfeld[10], Christopher Wood[7], Stuart L. Pimm[1,9]*

1 Nicholas School of the Environment, Duke University, Durham, North Carolina, United States of America, 2 IUCN, Gland, Switzerland, 3 World Agroforestry Center (ICRAF), University of the Philippines Los Baños, Los Baños, Laguna, The Philippines, 4 Institute for Marine & Antarctic Studies, University of Tasmania, Hobart, Tasmania, Australia, 5 BirdLife International, David Attenborough Building, Cambridge, United Kingdom, 6 Department of Zoology, University of Cambridge, Cambridge, United Kingdom, 7 Cornell Lab of Ornithology, Ithaca, New York, United States of America, 8 Department of Earth and Environment, Kimberly Green Latin American and Caribbean Center, Florida International University, Miami, Florida, United States of America, 9 Saving Nature, Durham, North Carolina, United States of America, 10 American Bird Conservancy, The Plains, Virginia, United States of America, 11 Environmental Research Centre, Duke Kunshan University, Kunshan, China, 12 Department of Environmental Studies, University of California Santa Cruz, Santa Cruz, California, United States of America

* stuartpimm@me.com

## Abstract

Accurate maps of species ranges are essential to inform conservation, but time-consuming to produce and update. Given the pace of change of knowledge about species distributions and shifts in ranges under climate change and land use, a need exists for timely mapping approaches that enable batch processing employing widely available data. We develop a systematic approach of batch-processing range maps and derived Area of Habitat maps for terrestrial bird species with published ranges below 125,000 km$^2$ in Central and South America. (Area of Habitat is the habitat available to a species within its range.) We combine existing range maps with the rapidly expanding crowd-sourced eBird data of presences and absences from frequently surveyed locations, plus readily accessible, high resolution satellite data on forest cover and elevation to map the Area of Habitat available to each species. Users can interrogate the maps produced to see details of the observations that contributed to the ranges. Previous estimates of Areas of Habitat were constrained within the published ranges and thus were, by definition, smaller—typically about 30%. This reflects how little habitat within suitable elevation ranges exists within the published ranges. Our results show that on average, Areas of Habitat are 12% *larger* than published ranges, reflecting the often-considerable extent that eBird records expand the known distributions of species. Interestingly, there are substantial differences between threatened and non-threatened species. Some 40% of Critically Endangered, 43% of Endangered, and 55% of Vulnerable species have Areas of Habitat larger than their published ranges, compared with 31% for Near

**Data Availability Statement:** The data is now hosted in an approved data repository (OSF) and

the updated link is provided in the
Acknowledgements: (https://osf.io/snmk4/).

**Funding:** The authors thank their home institutions
for providing support.

**Competing interests:** The authors have declared
that no competing interests exist. The views
expressed in this publication do not necessarily
reflect those of IUCN.

Threatened and Least Concern species. The important finding for conservation is that
threatened species are generally more widespread than previously estimated.

## Introduction

Species range maps play vital roles in ecology, biogeography, and conservation. Compiling
them takes considerable effort and it is time-consuming to update them in response to both
changing knowledge from new presence records and changing landcover and habitat availabil-
ity. We introduce a novel approach to producing range maps to address these difficulties, capi-
talizing on the increasing availability of high-resolution remote sensing data and abundant
crowd-sourced field observations. We illustrate our approach for the 1,151 terrestrial bird spe-
cies occurring from Mexico southwards through South America for which BirdLife Interna-
tional and Handbook of the Birds of the World [1] estimate resident plus breeding ranges
<125,000 km$^2$, and for which data are available. This region harbours the greatest bird diver-
sity globally [2], most of its species are forest dependent, and this range threshold encompasses
most species at risk of extinction due to habitat loss [2]. We examine how these new maps
compare with those published previously and provide examples of how they may inform
assessments of extinction risk. These new maps point to some species likely being more wide-
spread than previously thought, while others appear surprisingly limited in their likely distri-
bution. Importantly, our maps provide the underlying observations and GIS layers in a simple,
transparent manner and allow ready updates as data expand or are corrected and updated.

The IUCN Red List of Threatened Species, hereafter referred to as 'the IUCN Red List' [3],
requires a distribution map as part of each species' assessment. For terrestrial vertebrates, these
typically comprise polygons of range boundaries determined through expert interpretation of
published and unpublished distribution records. These records derive from museum speci-
mens, crowd-sourced initiatives, journal articles, distribution atlases, and unpublished reports,
and are combined with data on elevation, habitat preferences, and the distribution of those
habitats. Henceforth, for simplicity, we call the existing breeding plus resident range maps for
birds on the IUCN Red List [3] the "published ranges."

Using a set of five criteria, IUCN Red List assessments categorise species from Extinct
through Critically Endangered, Endangered, Vulnerable—which together are known as
'threatened'—to Near Threatened and Least Concern. These categories have quantitative
thresholds relating to rates of decline, range size and structure, population size and structure
and trends, and quantitative analysis of extinction probability [3]. One parameter that contrib-
utes to assessment is the "extent of occurrence", or "the area contained within the shortest con-
tinuous imaginary boundary which can be drawn to encompass all the known, inferred or
projected sites of present occurrence of a taxon, excluding cases of vagrancy" [3]. Recent work
has defined another representation of a species' distribution—the Area of Habitat or "the habi-
tat available to a species, that is, habitat within its range", which can also provide valuable
insight into a species' status [4]. One practical means of determining such an area is to refine a
species range map by its known elevation range and suitable landcover [5]. Recent studies
have suggested that Area of Habitat averages about 30% of the area of the published ranges for
birds, although wide variation exists in how much smaller Areas of Habitat are compared to
published ranges [5, 6].

Here, we develop a systematic approach of batch-processing range maps and deriving Area
of Habitat maps. The title of this paper reflects the numerous advances these methods entail.

First, dramatic changes have occurred recently in both the knowledge of species distributions as well as range shifts due to climate change and habitat loss. Therefore, mapping approaches that both enable batch processing and employ newly available data fill an important need.

Second, readily accessible, fine-scale global data now exist on relevant environmental variables, including elevation and land cover. Land cover is crucial in determining species distributions and yet human actions are rapidly shrinking [7] and fragmenting native habitats [8]. We focus on forest species because forest habitats can be readily distinguished from non-forested ones and because forests account for a high number of species. That said, these methods may be used with any landcover data of interest that can be matched to species' habitat preferences.

Third, we use a consistent analysis of observation localities. The rapid increase in crowd-sourced data both for birds—for example, in eBird (www.ebird.org)—and for other taxa—for example, in iNaturalist (www.inaturalist.org)—allows methods that can systematically and periodically incorporate locality data in the generation of maps. Crucially, eBird also allows inference of species' absences from non-detections. The often-expressed idea that "the absence of evidence is not evidence of absence" has obvious limits. Repeated visits to an area that consistently do not detect individuals may indeed provide evidence that a species is not present there [9]. Here, we consider 5km × 5km grid cells with ≥25 complete checklists without any presences of a given species as an indication of its absence. This interpretation is especially compelling given we only consider the subset of data comprising complete lists of all species recorded by an observer at a location.

Fourth, while understandable concerns exist about the quality of observational data, eBird has protocols in place to challenge and correct improbable observations [10]. A decision to exclude or include an outlying observation often requires knowing its provenance: who observed it, when, where, and under what circumstances, and how easy the species is to identify. Our protocol allows users to interrogate each datum and make their own decisions about its veracity. One strength of these methods is that one can easily re-run the models using only data in which the user has confidence.

This method for updating distribution maps using available landcover and observation data supports periodic reassessment of extinction risk as new datasets and observations become available, and thus facilitate timely conservation action.

## Methods and materials

### Initial filtering of species and data

Using birds as a model taxon for our methodology, we considered only terrestrial species, excluding those occupying marine or freshwater habitats. *A priori*, we excluded species that occur strictly in open habitats (see below) from our calculations of Area of Habitat. We identified these species from the habitat descriptions provided by www.birdlife.org and selected those for which the major habitats were grasslands, shrublands, and a few other open habitats. Most of these species occurred at high elevations and we estimated the thresholds of forest cover to be low.

We extracted the published range maps for all mainland bird species from Mexico southwards [11]. We included polygons with Presence codes of 1 (extant), Origin codes of 1 or 2 (native or reintroduced), and Season codes of 1or 2 (resident or breeding season ranges). All other polygons were excluded. Our initial target list consisted of the 1,228 species that have breeding plus resident ranges <125,000 km$^2$ (S1 Table). The 2019 taxonomy and Red List categories published in Handbook of the Birds of the World and BirdLife International [1] considered six of these species extinct or probably so, and eBird had no records of them. For 71

additional species, no qualifying eBird records existed (see below), so we excluded them from further analysis. Of these 71 species, 28 are threatened. An example of a non-threatened species with no qualifying records is the buff-breasted sabrewing, *Campylopterus duidae* (Least Concern), a poorly known, but apparently common hummingbird with a published range size of 102,777 km$^2$ in of the tepuis of Venezuela and northern Brazil. For 99 species, fewer than ten qualifying observations exist. Although these limited data produce a large scatter in estimates of Area of Habitat, we still included these species as these observations may inform map revisions that expand the published range of some species.

BirdLife International (and hence the IUCN Red List's) taxonomy [1] currently differs from eBird, which follows [12], and occasionally splits eBird's species into other species. It is possible to connect eBird observations to particular species in most of these cases because either the records have been entered in eBird for the specific taxon, or the relevant taxa are allopatric. For 29 species, however, this is not possible, as we discuss below.

After retrieving all approved and complete checklists for a species using eBird's '*auk*' package in R [13], we filtered out all checklists with Historical and Incidental protocols and those with a traveling protocol and travelling distance above 7 km or a duration of greater than 3 hours. (This follows eBird's rule for best practices [14].) We removed Historical and Incidental checklists because we lacked confidence in the accuracy of these points, and the traveling restrictions serve as threshold of confidence in the recorded location (see S1 Text for a more detailed discussion). We used the same criteria when retrieving checklists for calculating absences where the species was undetected.

We devised this filtering protocol based on extensive review of eBird data combined with expert opinion on what makes for plausible data. This is in addition to eBird's existing review process for removing false positives. Following this protocol results in the exclusion of the least plausible data and would allow new users to generate reasonable maps with confidence. This is not to say that this will generate the optimal map every time. Users may wish to make species-specific changes such as when this protocol is too conservative and removes legitimate observations in new areas.

## Producing the maps

We created an alpha hull around the union of the range map vertices and the eBird presence points (Fig 1A) using the '*alphahull*' package [15]. These alpha hulls "pinch in" the boundaries from a minimum convex polygon, thereby providing a more conservative range boundary. It is more tightly fitted to the underlying data, particularly where a species has a curved distribution, such as a species occurring along the Andes. The alpha value controls the degree of "pinching". Infinite alphas result in a shape identical to a minimum convex polygon while very small alphas result in a polygon for each point. For our methods, we use the median inter-presence distance as the alpha value for each species. Therefore, if the points are evenly distributed, the resulting alpha hull will be a single polygon. If the point data are clustered, the resulting alpha hull will split into multiple polygons.

To create an estimate of elevational range for each species, we calculated the distribution of elevations at the presence points using the GLOBE 1 km digital elevation model [16]. To restrict the impact of outliers, we excluded the highest and lowest 1% of the elevation estimates. Inspection of eBird data along with personal experience suggests that some reported locations may be a convenient landmark, such as a nearby peak, settlement, or lake, which can thereby bias the elevational range. We then compared the 1$^{st}$ and 99$^{th}$ percentiles with the documented elevation preferences in the IUCN Red List assessment. We used the smaller of the calculated GLOBE and documented Red List elevations as the minimum and the larger of the

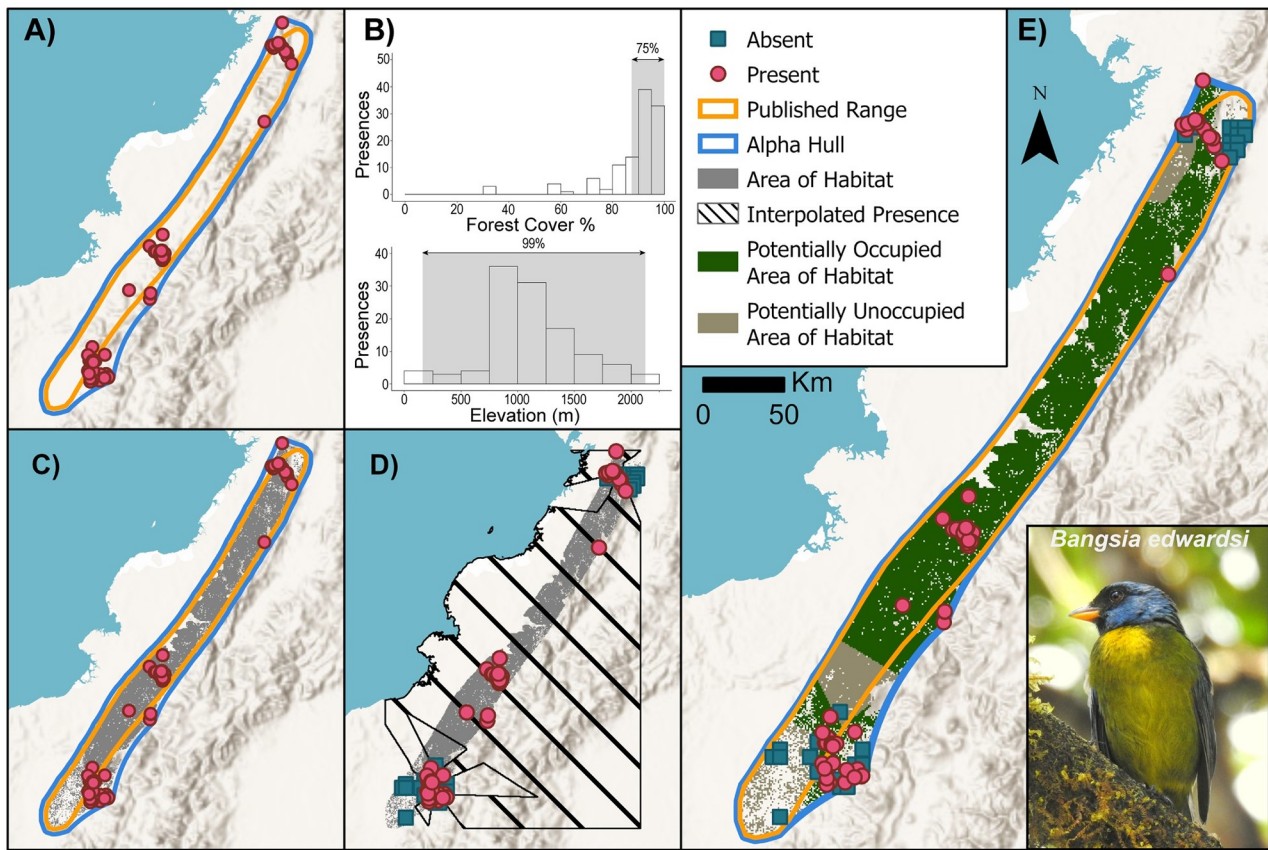

**Fig 1. Overview of steps in creating a range map for moss-backed tanager *Bangsia edwardsi*.** (A) After compiling presence locations (pink dots) and the current published range (orange lines), the user creates an alpha hull (blue lines) using the median inter-presence distance for alpha. (B) Users determine the habitat and elevational requirements for each species using distributions of landcover data extracted at each presence. (C) The next step is to refine the alpha hull area by the elevational and habitat land cover to identify the Area of Habitat (grey shading). (D) Using absence data (blue squares) generated from an aggregation of checklists that record no presences of the species, the user runs a nearest neighbour interpolation to identify all pixels that are closer to a presence or an absence. (E) The Area of Habitat is then split into Potentially Occupied Area of Habitat (green shading) and Potentially Unoccupied Area of Habitat (brown shading) based on the interpolation of the previous results. Base map provided by USGS and photo provided by Cristian Florez Pai with permission.

GLOBE and Red List elevations as the maximum to determine the final elevational range of the species.

We created the habitat layer using the 30 m Hansen et al. forest cover dataset [17] bilinearly resampled to 1 km. We extracted the tree cover percentage for every observation from 2000 onward within a 1 km × 1km cell. We excluded all occurrences falling within 0% of tree cover as these observations weighted negatively for forest species and often represented lodges or trailheads rather than the specific location of observation. Using the distribution of tree cover values, we defined the habitat range as the upper 75% of the observed values. This criterion excluded unusually low values that often represent settlements below the actual location of the bird (Fig 1B). It should also be noted that the Hansen et al. forest cover data include plantations, which may be unsuitable habitat for some species. Unsurprisingly, our estimates of Area of Habitat are only as good as the quality of the habitat data. Future efforts could incorporate more accurate habitat data or utilise data on other habitat types to map more species.

We calculate Area of Habitat from the intersection between the elevational and forest layers within the alpha hull (Fig 1C). We then refined the Area of Habitat into Potentially Occupied

Area of Habitat and Potentially Unoccupied Area of Habitat using a nearest-neighbour interpolation between presences and absences. We define absences as 5km × 5km grid cells that are surveyed with 25 or more complete checklists and no presences of the species of concern within the cell. When calculating these absences, we filtered the checklist data to keep only those that fall within pixels of the Area of Habitat. Thus, if a checklist falls in a location that is deemed to be inappropriate in terms of elevation or land cover, it is not counted towards the threshold for being absent, as one would not expect to find the species in unsuitable habitat. This choice differs from what we defined for presences, where all were retained even if they fall outside what we previously calculated as Area of Habitat. This was to allow for some approximation of recording a presence while we do not expect species to be recorded in checklists in unsuitable habitat. We used the centre of each absence cell for the interpolation (Fig 1D and 1E). For the final step, we compiled all layers and created the interactive HTML maps using the 'leaflet' package [18].

### eBird Historical and Incidental, and GBIF

For comparison purposes and to evaluate the potential of incorporating more data, we included Historical and Incidental eBird records as well as data from Global Biodiversity Information Facility (GBIF, www.gbif.org), which include observations from iNaturalist (www. inaturalist.org). These data are displayed on the interactive maps, though we do not calculate ranges using these data. For GBIF data, we applied a strict filtering process to remove all locations with issues associated with geolocation and missing coordinates. We retained records with "Rounded Coordinate" since that did not impact our confidence in those points. We removed all eBird records from the GBIF dataset to avoid redundancy.

## Results

We mapped the Area of Habitat for the 1,151 terrestrial bird species that have breeding and resident ranges from Mexico southwards. Of these, we are not confident in our estimates for 99 species that had fewer than 10 observation and for 92 desert, grassland, and shrubland species, given challenges in mapping these habitats.

Fig 2 shows an exemplar map for a hummingbird, the glittering starfrontlet, *Coeligena orina* (Endangered). This species has a small range in the Western Andes of Colombia, west of Medellin. We map Area of Habitat, dividing it into Potentially Occupied Area of Habitat in dark green—these are cells nearer to sightings of the species—and Potentially Unoccupied Area of Habitat in brown—cells nearer to where observers have repeatedly not recorded the species. Some of the latter areas are consistently well surveyed without recording a presence, while others may represent a call for more exploration.

Unsurprisingly, estimates of Area of Habitat and Potentially Occupied Area of Habitat correlate positively with the extent of published ranges (Fig 3). As one might expect, estimates based on fewer than ten observations often relate to species with small ranges. They show a wider scatter, reflecting the small sample size. The outliers—either those with larger areas of habitat or smaller—suggest that re-evaluation of published ranges is warranted.

Although correlated, the relationship between the area of published ranges and our estimates of Area of Habitat varies widely (Fig 4). Across all forest species analysed, our estimates of Area of Habitat were on average 12% larger (σ = 185%) than the published ranges (S1 Table). Our estimates of Potentially Occupied Area of Habitat were on average 70% of the areas of published range (σ = 93%). Among threatened species (Vulnerable, Endangered, and Critically Endangered, n = 208), 24% have Areas of Habitat <50% of the published ranges due to lack of suitable habitat within the range. These proportions of species with Area of Habitat

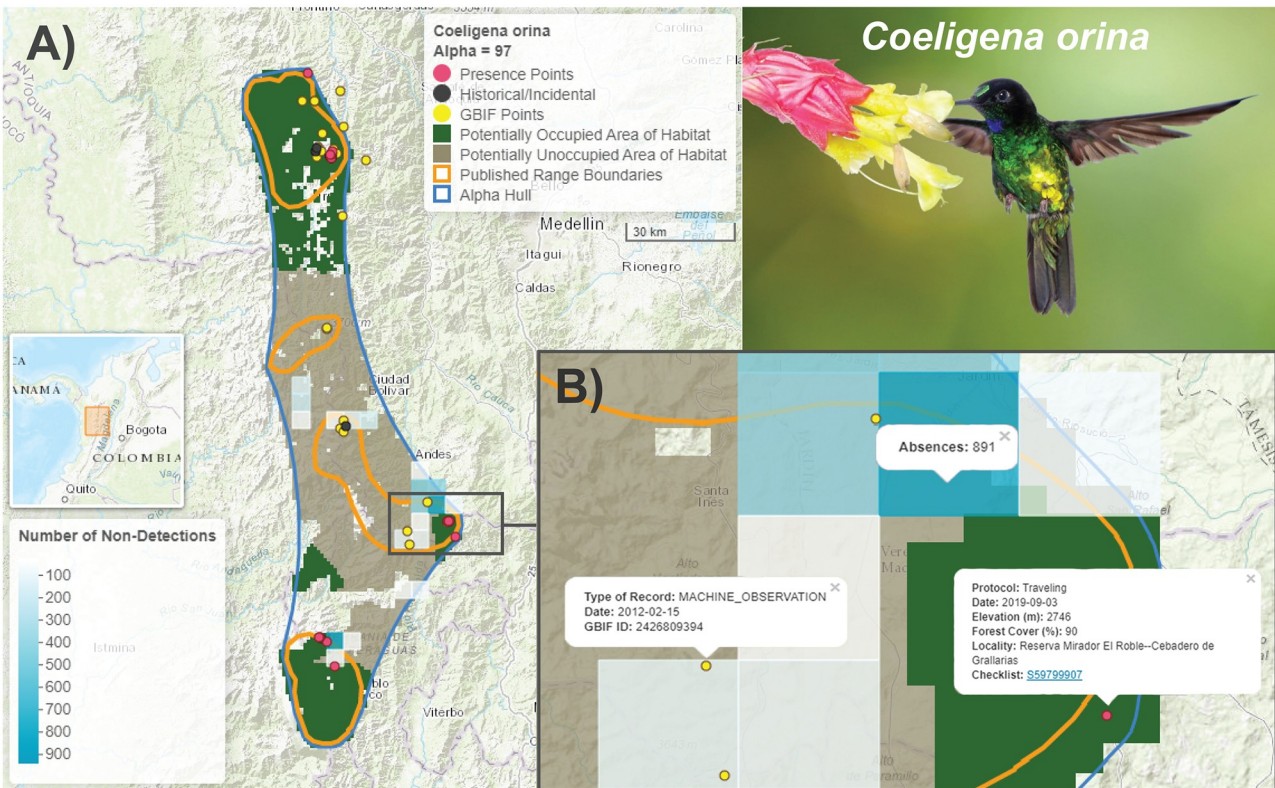

**Fig 2. Example interactive range map for the glittering starfrontlet, *Coeligena orina*.** (A) A screenshot of the interactive HTML map created with the 'leaflet' package that shows the alpha hull (blue) generated from the union of the published range maps (orange polygons, [11]) and the eBird observations (pink dots). The shaded areas are the Area of Habitat, i.e. the alpha hull refined by elevational and canopy cover requirements. Potentially Occupied Area of Habitat (shaded green) represents areas closer to observations than absences (absences are defined as the aggregate of 25 or more checklists within a 5km × 5km cell that do not record the species, shaded in blue, see Methods). In contrast, Potentially Unoccupied Area of Habitat (shaded brown) represents areas closer to absences than observations. Also shown on the map are presence localities from the Global Biodiversity Information Facility (yellow dots) and historical/incidental records from eBird (black dots) that were excluded from the analysis due to inconsistent quality, but which may provide additional insights. (B) The interactive nature of these maps allows users to zoom in on details and interrogate the data by clicking on any point to see the underlying metadata such as date, elevation and forest cover, and links to the original eBird checklists, or by clicking on an absence square to see the number of non-detections. Basemap provided by ESRI and photo by Luis Mazariegos with permission.

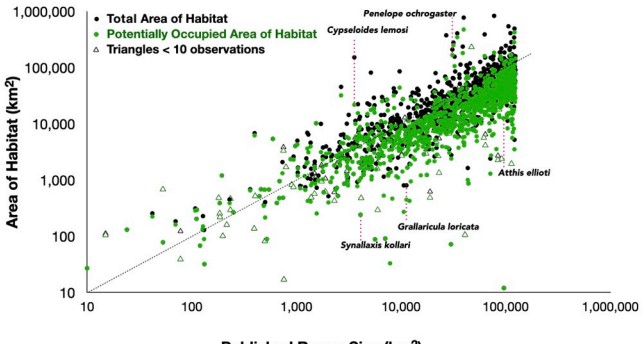

**Fig 3. Area of Habitat and Potentially Occupied Area of Habitat versus the published range.** Black symbols represent estimates of total Area of Habitat and green symbols represent Potentially Occupied Area of Habitat. Solid symbols represent species with ten or more observations, open triangles represent those with fewer than ten. The text discusses the five examples labelled. Dotted diagonal line shows identical values along the two axes.

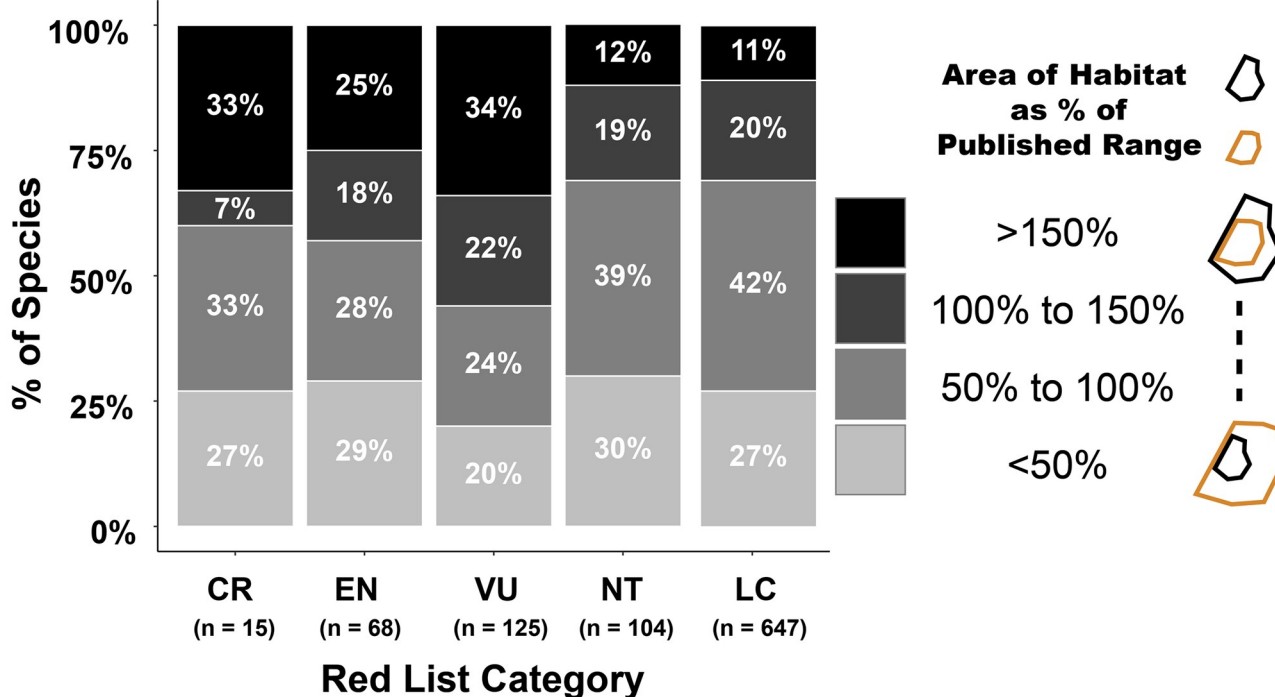

**Fig 4. Area of Habitat estimates in relation to published range sizes for species in different IUCN Red List categories.** Bars show the proportions of Critically Endangered (CR), Endangered (EN), Vulnerable (VU), Near Threatened (NT), and Least Concern (LC) species with different percentage values for Area of Habitat divided by the published range. The lightest colours represent species with Area of Habitat that is less than 50% of the published range while the darkest colours represent species with Area of Habitat more than 150% larger than the published range size. Black polygons represent Area of Habitat and gold polygons represent published ranges.

<50% of range are similar for Near Threatened (30%) and Least Concern (27%) species. Meanwhile, 31% of threatened species have Areas of Habitat >150% of the published ranges because eBird observations expand the range boundary beyond the published ranges. By contrast, the equivalent proportions are much smaller for Near Threatened (12%) and Least Concern (11%) species with Area of Habitat >150% of the published ranges. Such instances suggest that the species may be more widespread than previously recognised. Our approach facilitates checking of observations that expand known range and encourages their confirmation.

For some species, the published ranges include locations in which there are no eBird data (neither recorded presences nor absences inferred from complete checklists omitting the species). These may derive from information from remote areas that are inaccessible to most eBird observers (e.g., owing to current security concerns) or from areas that historical ornithological expeditions visited. These cases call for further research and exploration of the areas in search of the species. Alternatively, they may represent errors in the boundaries of published ranges or unoccupied areas within the species' distributional limits. (The range maps represent range boundaries rather than occupancy.) In many cases, we found areas within the alpha hull boundary (whether within the published range or not), where there are no records, but which nonetheless have forest cover within the elevation range that may support the species.

### Species for which Area of Habitat is greater than published range

Crowd-sourced data expanded the range boundaries of some species considerably (Fig 3). An extreme example is the white-chested swift, *Cypseloides lemosi* (Vulnerable). Known to science

only since 1962 and then from the upper Cauca Valley in Central Colombia—which the published range represents—eBird records show it to occur widely in Colombia, Ecuador, Peru, and into Bolivia. For this species, we estimated an Area of Habitat that is >40 times the published range size.

A second example is the chestnut-bellied guan, *Penelope ochrogaster* (Vulnerable) (Figs 3 and 5). The published range represents ten or so scattered locations in south-east Amazonia, while eBird records indicate a more continuous distribution.

We analysed 208 species listed as threatened. For 64 of these, our estimates of the Area of Habitat are >150% of the published ranges (Fig 4 and S1 Table). That crowd-sourced data considerably expand these published ranges need not necessarily change their Red List category. For example, some large-ranged species are listed as threatened owing to rapid rates of decline. Examples include various parrot species trapped for the pet trade and guan and curassow species that are heavily hunted. Other species with large published ranges may have small populations because their ranges are exceptionally fragmented [19, 20], or because the species occurs at naturally low densities. Nonetheless, some of these species may prove to be less threatened than currently considered.

## Species for which Area of Habitat is smaller than published range

Of potential conservation concern are those species that have an Area of Habitat much smaller than the published range. A disadvantage of using polygons to map species distributions is that they mask areas lacking habitat within the range boundaries, often prevalent in regions experiencing rapid land cover changes. Lack of understanding of habitat distribution may lead to over-estimating population sizes (or underestimating decline rates) and hence underestimating the extinction risk of such species.

An extreme example is the hoary-throated spinetail, *Synallaxis kollari* (Critically Endangered) which has an Area of Habitat of only 6% of the published range (Fig 3 and S1 Table). It lives in narrow gallery forests along rivers in northern Brazil that transit areas that are otherwise grasslands [21]. Agricultural expansion threatens the remaining habitat. Dependence on forests explains its small Area of Habitat and points to the limitation of creating simple polygons to describe ranges. Only a much finer, more convoluted polygon would describe this range more accurately.

Two more examples are from Central America: the wine-throated hummingbird, *Atthis ellioti* (Least Concern) (Figs 3 and 6), and the black-throated jay, *Cyanolyca pumilio* (Least Concern). Both are listed as Least Concern because of their large Extent of Occurrence and inferred population sizes and lack of rapid population declines. The hummingbird has a published range of 94,985 km$^2$. eBird observations extend beyond its published range substantially south to mountains in El Salvador and well to the east in Honduras, yet we estimate the species' Area of Habitat at only 25,804 km$^2$, falling to 12,594 km$^2$ if we consider only Potentially Occupied Area of Habitat. The black-throated jay resides in a highly fragmented landscape [22]. We estimate its Area of Habitat as 21,421 km$^2$ while its published range is 103,160 km$^2$. While the Extent of Occurrence for both species exceed the thresholds to trigger assessment as a threatened species, extensive and continuing deforestation leaves the remaining forests where these species occur in widely isolated fragments. These facts suggest the importance of careful monitoring of possible threats to these species is needed.

A final example is the scallop-breasted antpitta, *Grallaricula loricata* (Near Threatened) (Figs 3 and 7). We estimate an Area of Habitat to be only 447 km$^2$ in several widely separated forest patches. The new mapped distribution of this species may inform future reassessment of whether more than half the population occurs in small and isolated subpopulations (i.e.

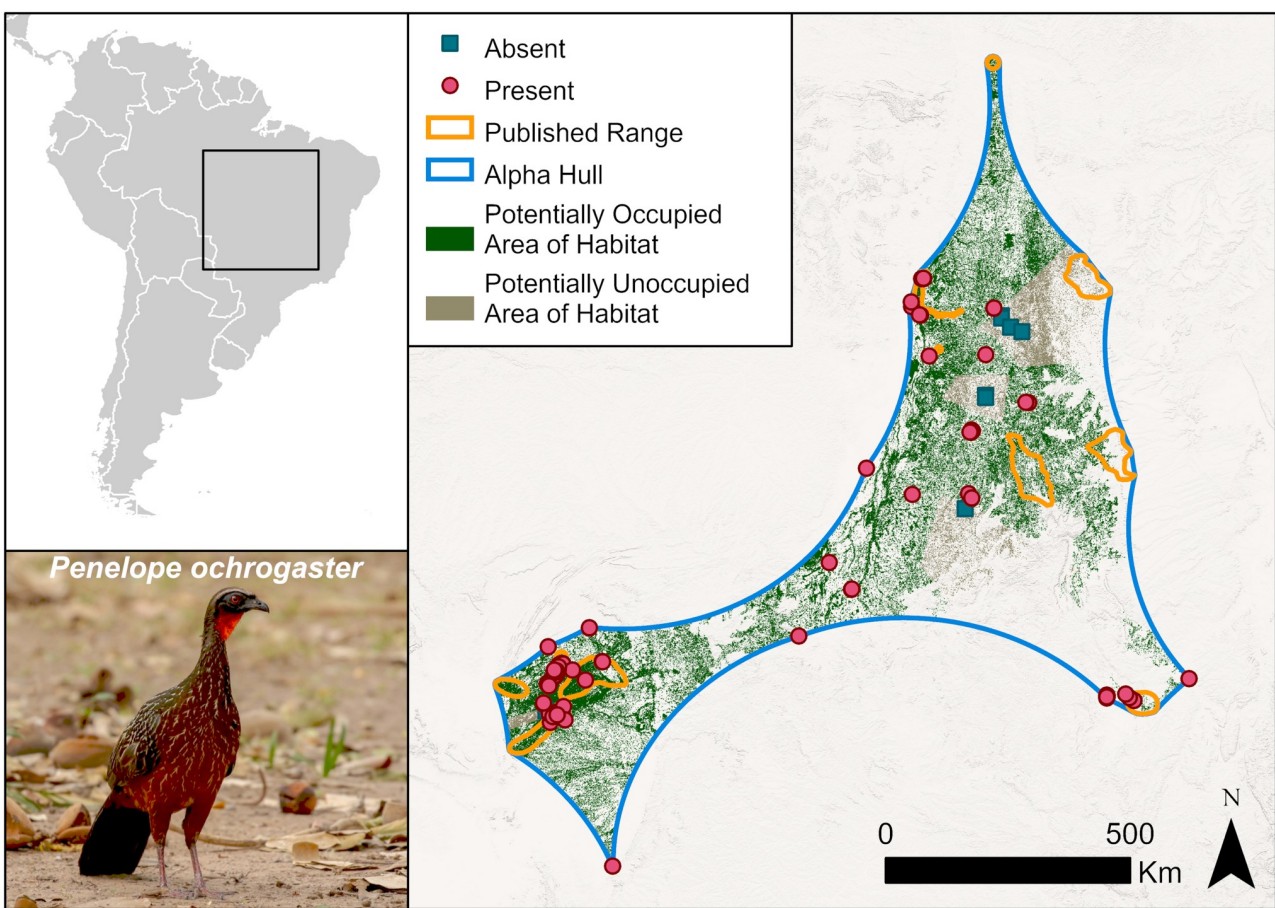

**Fig 5. Distribution of the chestnut-bellied guan, *Penelope ochrogaster*.** eBird records illustrate that the range of *P. ochrogaster* expands beyond the published range. Symbology is simplified to show absences (blues squares) as the centroids of absence grid cells. Base map provided by USGS and photo by Carlos Sanchez with permission.

"severely fragmented") or within fewer than 10 locations (as defined by threats, following the definitions of IUCN [3].

## Discussion

### General findings

Previous estimates of Areas of Habitat were constrained within the published ranges and thus were, by definition, smaller—typically about 30% [6]. This reflects how little habitat within suitable elevation ranges exists within those published ranges for the species. Our results show that on average, Areas of Habitat are 12% *larger* than published ranges, reflecting the often-considerable extent that eBird records expand the known distributions of species beyond published ranges.

Interestingly, there are substantial differences between threatened and non-threatened species. Some 40% of Critically Endangered, 43% of Endangered, and 55% of Vulnerable species have Areas of Habitat larger than their published ranges, compared with 31% for Near Threatened and Least Concern species (Fig 4). This may be because non-threatened species are better known, more common, and have generally larger ranges than the threatened species. Alternatively, threatened species, particularly those with small ranges, may have been mapped at finer

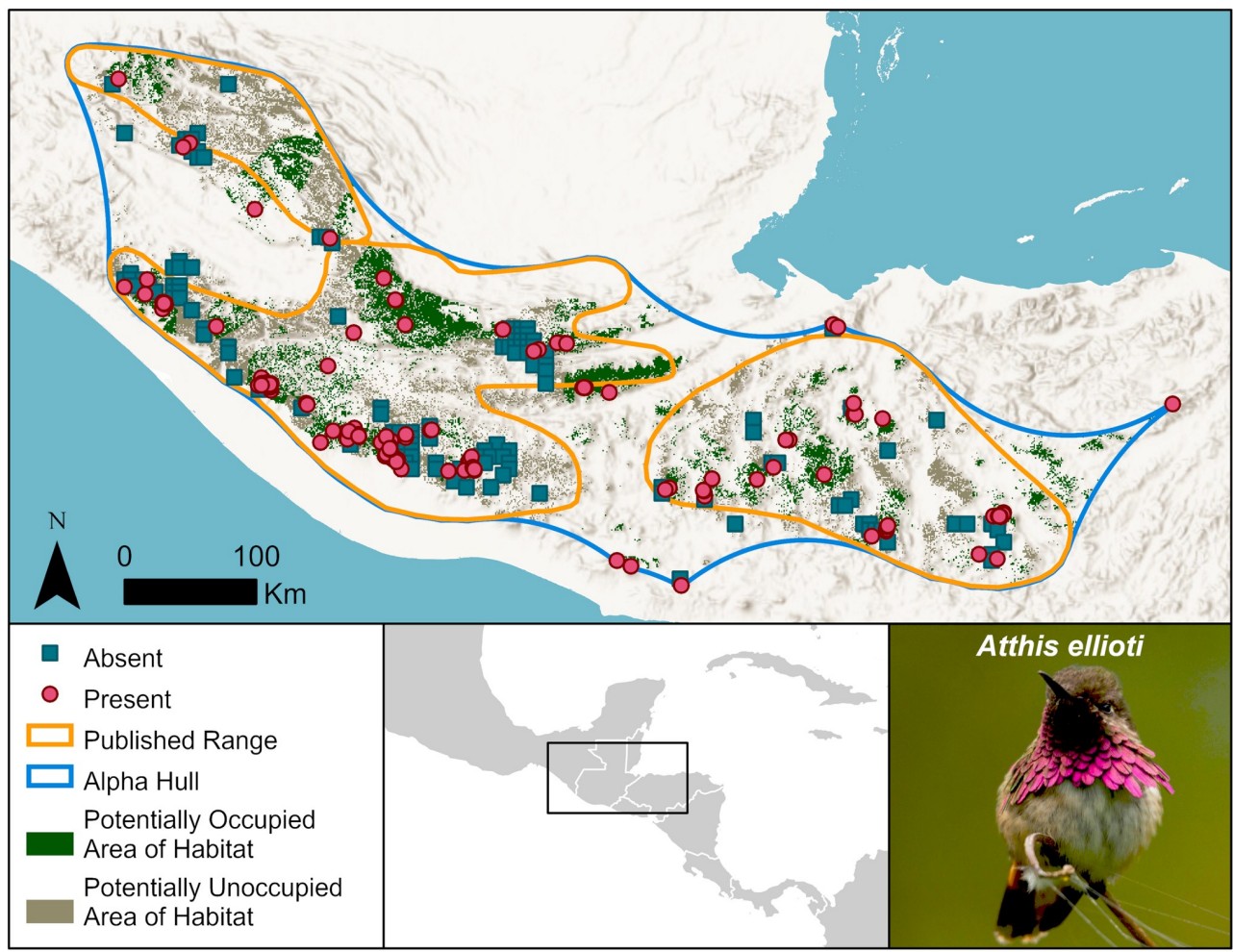

**Fig 6. Distribution of the wine-throated hummingbird, *Atthis ellioti*.** The published range boundaries are large, but extensive deforestation has left the remaining forests few and fragmented, resulting in a much smaller Area of Habitat. Symbology is simplified to show absences (blues squares) as the centroids of absence grid cells. Basemap provided by USGS and photo provided by David Mora Vargas (Macaulay Library, ML296645721).

resolution than non-threatened species, i.e. their range boundaries have been delineated to exclude areas in which the species in absent, whereas those of non-threatened species are cruder approximations of the range limits. The important finding for conservation is that threatened species are generally more widespread than previously estimated.

## Advantages

**Transparency.** The maps are explicit about what we know and what we do not. They predict Areas of Habitat where we expect the species to live but in which observers have not recorded it in eBird. They also predict areas within existing range maps where a species is unlikely to live because the elevations or habitat are outside its known preferences. For instance, in some areas within Central America (Fig 6), what habitat remains is massively fragmented, posing additional risk to species over and above the simple habitat loss [19, 20]. eBird's data collection protocols, which by requiring observers to document whether or not every bird recorded is included in a checklist, allow assessment of absence and presence, and thus innovation to differentiate between Potentially Occupied Area of Habitat and Potentially

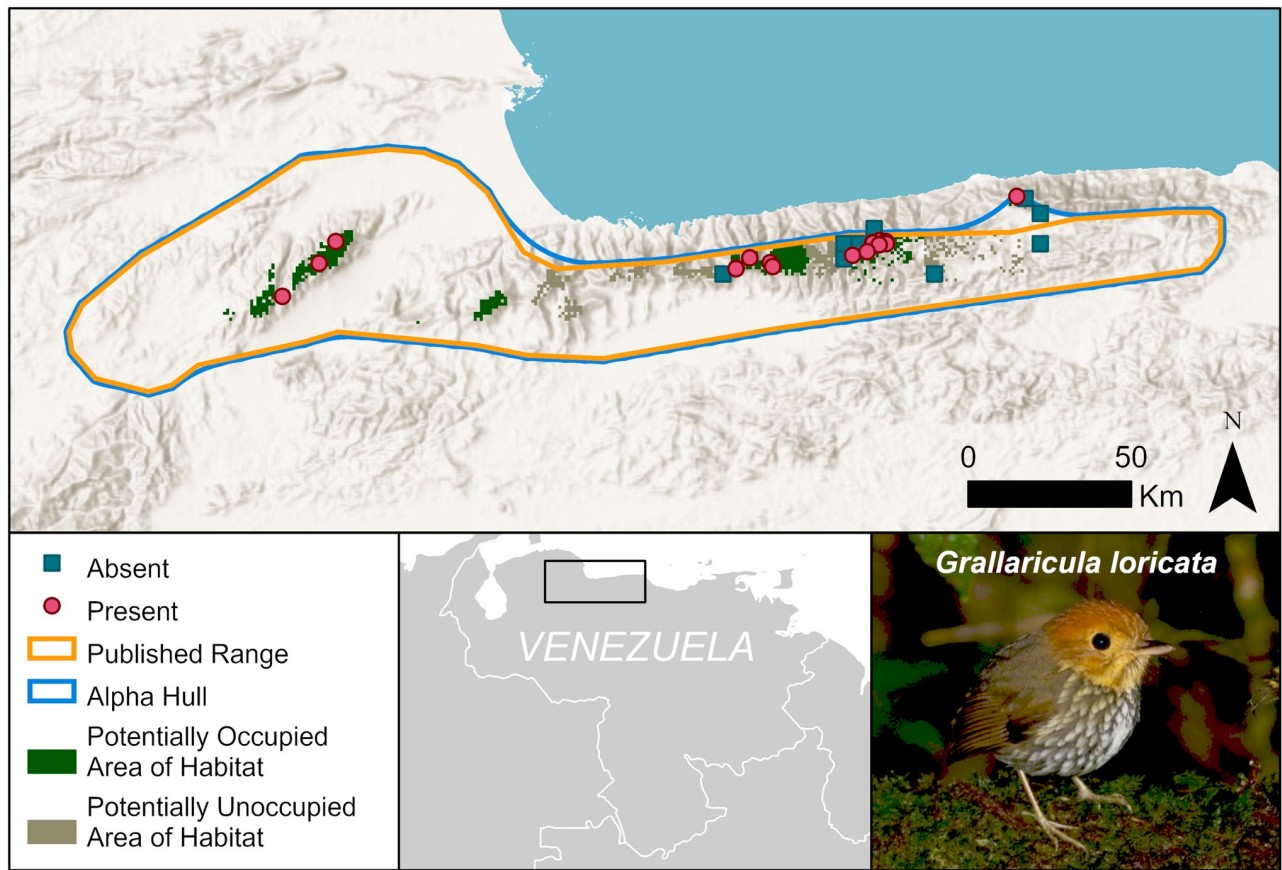

**Fig 7. Distribution of the scallop-breasted antpitta, *Grallaricula loricata*.** Due to habitat loss and restricted elevational range, the Area of Habitat is much smaller than the current published range size. Symbology is simplified to show absences (blues squares) as the centroids of absence grid cells. Basemap provided by USGS and photo provided by Margareta Wieser (Macaulay Library, ML205392301).

Unoccupied Area of Habitat. Development and application of similar techniques for assessing absence using other crowd-sourced species datasets, such as iNaturalist, would be necessary to enable the extension of this innovation to taxa other than birds.

Our maps predict where further exploration would be prudent. Birdwatchers tend to visit known sites for particularly rare species, rather than explore new ones, to the detriment of modelling species ranges [23]. (Of course, in seeking rarities, they also document other species coincidentally.) Our maps also predict where the species might occur, but nearby surveys have not found it, thereby hopefully encouraging further exploration.

**Provenance.** Interactive maps permit users to interrogate the provenance of observations by querying individual records to check their likely validity. Without provenance, we cannot understand how maps are produced and what might be their limitations. Users can interpret that information as they see fit.

**Revision.** By design, we developed the methods herein to enable users to update maps at regular intervals as observations accumulate and forest cover changes. The methods can include other sources of records. We can explore how eBird data availability alters our maps, either through growth in the dataset over time or through analysing subsets or supersets of the data.

In addition, as further context, our maps display data from GBIF and historical observations from eBird that we did not use to calculate Areas of Habitat. Inspection of those data

may suggest locations where species are unlikely to occur now because no habitat remains. Our maps afford the chance to study the changes in Areas of Habitat that have occurred and will likely occur in the future.

## Limitations

**High elevation, desert, and other open land species.** Given the habitat data we used, we were not confident in mapping Area of Habitat for 92 non-forest species (S1 Table). We checked these individually against the species' known elevations and habitat preferences and excluded them from the statistical summaries. Nonetheless, eBird observations may still yet expand the species' range beyond what is currently published. One such example is the white-bellied cinclodes, *Cinclodes palliatus*, an open area species that is Critically Endangered owing to small and declining populations. For such species, we can produce maps limited by elevation, but not habitat (Fig 8). These elevation-only maps often appear very similar to the published ranges. Although accessible, high-resolution vegetation datasets are predominantly available for forest habitat, our method has the potential to be expanded to other habitats when appropriate datasets become available. At present, we have not extended our approach to other habitats by matching landcover classes at points where eBird observations fall to habitat classes as coded on the IUCN Red List owing to errors of omission and commission in such matches.

**Other habitat specialists.** Some species specialise on narrow or monodominant vegetation types within forest but not distinguished by our forest cover data. Our Area of Habitat estimates are likely too extensive for these species, and additional analyses will be needed to estimate them more precisely (see S1 Text for a more detailed discussion). These include many bamboo habitat specialists [24–26], some of which are nomadic, depending on periodic mass seeding events, a strategy that may have doomed the purple-winged ground dove, *Paraclaravis geoffroyi* (Critically Endangered), to extinction. Its published range is nearly half a million square kilometres—the range over which it searched for seeding bamboo.

Some 7% of the Amazon Basin consists of white-sand ecosystems [27]. Together with other forest types limited to nutrient-poor soils, this ecosystem harbours many habitat specialists. Other species within the Amazon Basin are restricted to floodplain forests; examples include the wattled curassow, *Crax globulosa* (Endangered), and the pearly-breasted conebill, *Conirostrum margaritae* (Vulnerable). Mangrove specialists such as the mangrove hummingbird, *Amazilia boucardi* (Endangered), may pose a challenge because some landcover datasets do not differentiate mangroves and adjacent moist forests [28, 29].

**Intra-tropical migratory birds.** Although most of the species we analyse are mainly resident birds, some species (e.g., swifts, hummingbirds, and songbirds) make regular but poorly understood movements and migrations across altitudinal and wet-dry gradients. Likewise, some species are more prone to vagrancy than others, including Andean species that might occasionally descend to the lower elevations or even the lowlands as a temporary escape from harsh weather. Different mapping methods are required to distinguish breeding habitat from other habitats for these species.

**Introduced species.** eBird records include species outside their native range. For example, most eBird records of the red-crowned amazon, *Amazona viridigenalis* (Endangered) are from southern California, where it is introduced, and southern Texas, where it may be a colonist. Fewer observations exist in its known native range in Mexico, a fact evident for this species, but not always so for other introduced species.

**Taxonomic issues.** eBird and BirdLife International are working towards, but have not yet completed, a common taxonomy. Even a taxon as well-known as birds generates

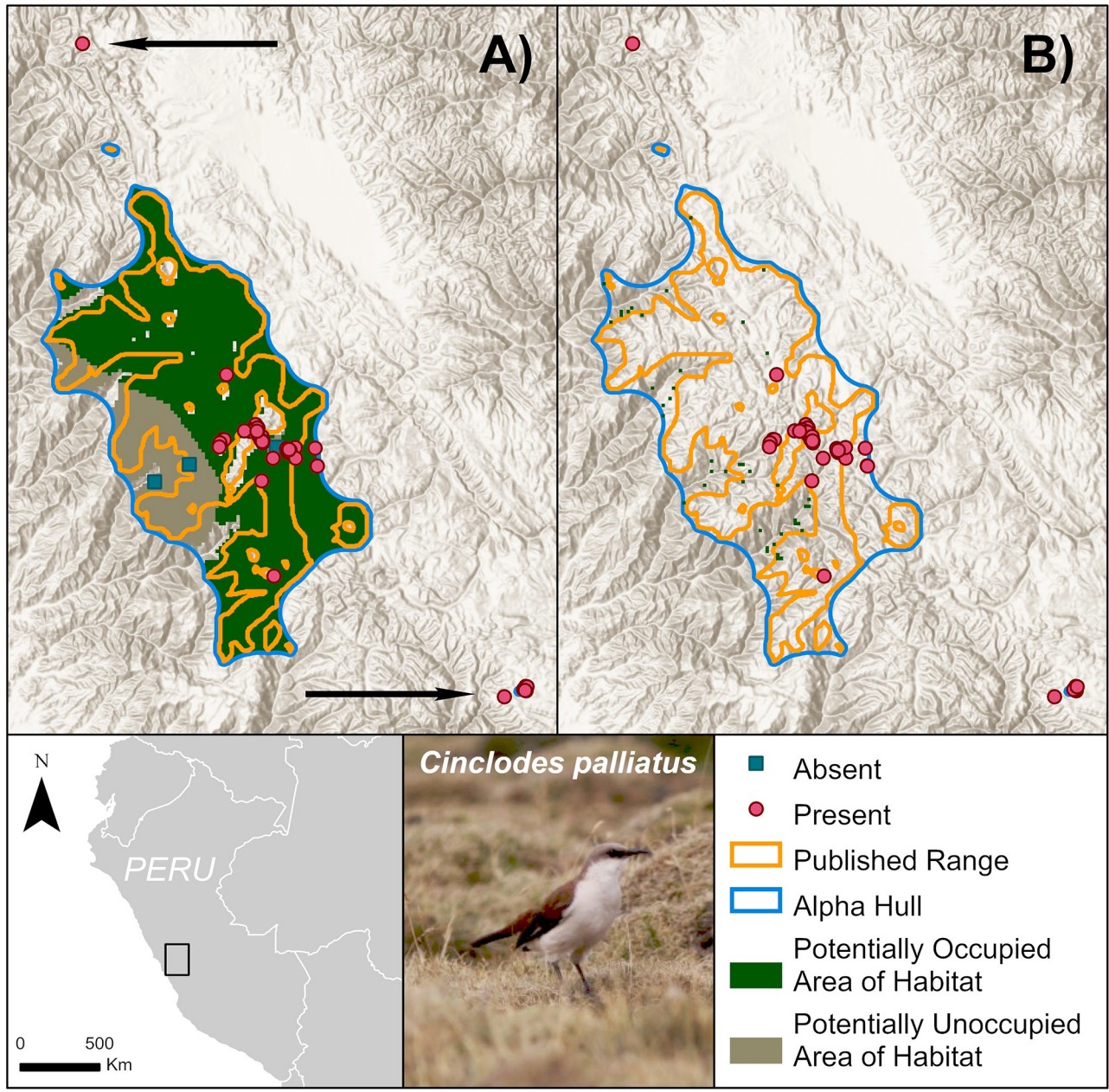

**Fig 8. Limitations of refining Area of Habitat for white-bellied cinclodes, *Cinclodes palliatus*.** (A) Estimates of Area of Habitat for species that prefer open land are often similar to the published ranges when refined just by the estimated elevational ranges. (B) Further refining a species range using forest cover—not a preferred habitat—may lead to a significant underestimation of Area of Habitat. Despite lack of appropriate landcover data to refine Area of Habitat, eBird records beyond the published range (identified with arrows) inform us that the range of these species is wider than recognised. Symbology is simplified to show absences (blues squares) as the centroids of absence grid cells. Basemaps provided by USGS and photos provided by Stuart Pimm with permission.

taxonomic debates, often prompted by improving knowledge. Debates among taxonomists will never disappear entirely, but the maps we produce can, nonetheless, provide useful insights into the issues.

Some differences merely involve different generic names. For 135 species in our sample, however, the issue is that Handbook of the Birds of the and BirdLife International [1] split out

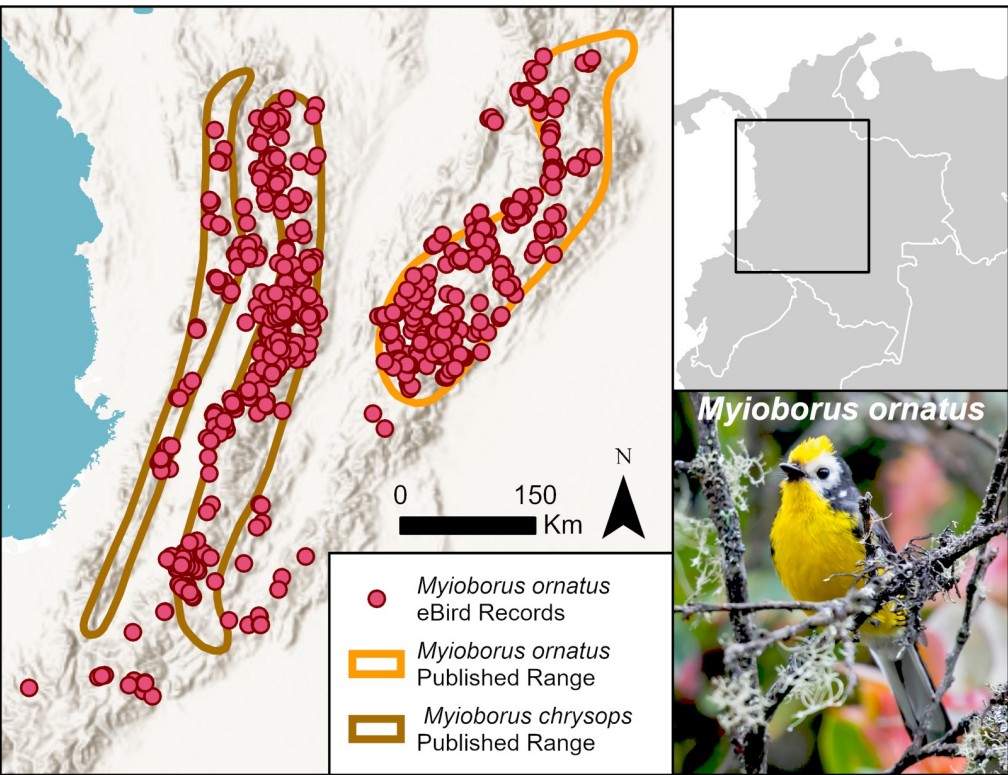

**Fig 9. eBird observations of the *Myioborus ornatus* superspecies.** eBird considers *Myioborus ornatus* to be one species, but [1] treat it as two separate species: *Myioborus ornatus* in the Eastern Andes (orange polygon) and *Myioborus chrysops* in the Central and Western Andes (brown polygon). Basemap provided by USGS and photo provided by Juan José Arango with permission.

what eBird considers to be subspecies. Many occur in the Andes where the topography isolates montane taxa into separate populations. An example is the violet-throated starfrontlet, *Coeligena violifer* (Least Concern), which eBird considers to be one species but [1] treats as four separate species, splitting out C. *dichroura* (Least Concern), *C. albicaudata* (Least Concern), and C. *osculans* (Least Concern).

This easy separation is not always the case, whereupon eBird observations afford insights for how taxonomists might draw their conclusions. An example is the golden-fronted redstart, which [1] considers to be two species: yellow fronted redstart, *Myioborus ornatus* (Least Concern), and golden-fronted redstart, *Myioborus chrysops* (Least Concern) (Fig 9). The numerous eBird observations cluster into three separate mountain ranges in Colombia—the Western, Central, and Eastern Andes, but extend much further south in the last two ranges than the published ranges suggest. Analysis of the genetics of this species group would likely be informative.

Of greatest concern from a conservation standpoint are species that, when split, are assessed as threatened on the IUCN Red List. Due to difficulties in assigning eBird records to the appropriate taxon, we were unable to produce maps for two such threatened species: the lilacine amazon, *Amazona lilacina* (Critically Endangered), and the white-tailed canastero, *Asthenes usheri* (Vulnerable).

**Methodological decisions.** Our methods include various decisions, including how we filter observations, the use of alpha-hulls and their parameterization, the selection of elevational limits, the grain of the elevation maps, the forest cover limits and others. All are arbitrary and

the choices we make are motivated by pragmatism. We explored some of their consequences in the supplementary materials. Estimates of the elevational range for instance, may be dependent on the DEM resolution one uses in conjunction with eBird data. Therefore, taking the union with previously published range estimates is the more conservative solution. Importantly, because our methods are transparent and the maps we produce allow one to interrogate each datum, one can readily explore the consequences of the decisions we have made.

## Conclusions

We present a novel and transparent approach to mapping and updating range maps for birds using crowd-sourced observation data and spatial elevation and land cover information. These maps, and the method used to produce them, are freely available, so that the maps can be periodically updated based on new observation and habitat data. The utility of these maps is broad, providing a basis for potentially evaluating species' natural history, extinction risk, habitat threats, and conservation priorities.

This paper's authors include representatives from the IUCN and BirdLife International (Red List Authority for birds, and hence responsible for their Red List assessments, including the production of maps of current ranges). Authors from American Bird Conservancy (www. abcbirds.org) and Saving Nature (www.savingnature.org) help acquire land for protection based on where threatened species live, hence depend on the best possible maps to identify priorities. Also represented is the Cornell Lab of Ornithology eBird team (www.ebird.org), an organization that has archived a billion crowd-sourced observations of birds globally. Lastly, included is a community of scientific users interested in bringing the rapidly expanding GIS data to bear on species mapping problems.

The maps produced inform the needs of these various constituencies, being transparent in the provenance of their data, straightforward to update when needed, and feasibly aggregated to identify areas of conservation concern and, as above, areas where further exploration would be helpful.

## Supporting information

Each species' mapped elevation ranges, forest cover thresholds, major habitat requirements, Area of Habitat estimates, published range sizes, and IUCN Red List category are given in S1 Table. The supplementary materials provides an extended technical description of methodology for combining published data on geographical ranges, elevations, and habitat preferences with crowd-sourced data. S2 Table provides a detail set of estimated elevational ranges of species. Additionally, we supply code in R for the algorithm described in the Methods.

**S1 Text. Analysis and discussion of methodological choices on results.**
(DOCX)

**S1 Code. Detailed description and full code base for reproducing methods.**
(PDF)

**S1 Table. Summary table of species analyzed.** Table detailing all target species identified, reason for exclusion from the analysis (if any) and all relevant metrics.
(XLSX)

**S2 Table. Summary table of elevational estimates.** Table of elevational estimates derived from BirdLife, Stotz et al. (1996), or from crowd-source data overlaid on either 90m or 1km DEM.
(XLSX)

**S3 Table. Effects of elevational estimates on Area of Habitat.** Comparison of using different estimates of elevational range in calculating Area of Habitat for the eight species with the largest discrepancies.
(XLSX)

## Acknowledgments

We would like to thank the community of eBird users who have contributed data, without which this work would not be possible, and the large number of contributors to BirdLife's Red List assessments (including range maps) for all birds. The first generation of maps may be viewed at https://osf.io/snmk4/.

## Author Contributions

**Conceptualization:** Ryan M. Huang, Wilderson Medina, Clinton N. Jenkins, Binbin V. Li, Natalia Ocampo-Peñuela, Stuart L. Pimm.

**Data curation:** Ryan M. Huang, Wilderson Medina.

**Formal analysis:** Ryan M. Huang, Wilderson Medina.

**Funding acquisition:** Ryan M. Huang, Daniel J. Lebbin, Stuart L. Pimm.

**Investigation:** Ryan M. Huang, Wilderson Medina.

**Methodology:** Ryan M. Huang, Wilderson Medina, Clinton N. Jenkins, Alison Johnston, Natalia Ocampo-Peñuela, Hannah Wheatley, David A. Wiedenfeld.

**Project administration:** Ryan M. Huang, Wilderson Medina, Stuart L. Pimm.

**Resources:** Ryan M. Huang, Wilderson Medina.

**Software:** Ryan M. Huang, Wilderson Medina.

**Supervision:** Ryan M. Huang, Stuart L. Pimm.

**Validation:** Ryan M. Huang, John W. Fitzpatrick.

**Visualization:** Ryan M. Huang.

**Writing – original draft:** Ryan M. Huang, Stuart L. Pimm.

**Writing – review & editing:** Ryan M. Huang, Wilderson Medina, Thomas M. Brooks, Stuart H. M. Butchart, John W. Fitzpatrick, Claudia Hermes, Clinton N. Jenkins, Alison Johnston, Daniel J. Lebbin, Binbin V. Li, Natalia Ocampo-Peñuela, Mike Parr, Hannah Wheatley, David A. Wiedenfeld, Christopher Wood, Stuart L. Pimm.

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
