## [Decision Letter · Decision Letter 0]

28 Sep 2021

PONE-D-21-23994Batch-produced, GIS-informed range maps for birds based on provenanced, crowd-sourced data inform conservation assessmentsPLOS ONE

Dear Stuart,

 At long last, both reviews are in! Congratulations! Both the reviewers and I strongly feel this represents a very valuable (and useful) addition to the literature.  However, there were a few small suggestions that probably should be tended to. Hence the "minor revision". But these are largely cosmetic and should not cause too much grief.  Therefore, we invite you to submit a revised version of the manuscript that addresses the points raised during the review process.

We look forward to receiving your revised manuscript.

Kind regards,

Tim A. Mousseau

Academic Editor

PLOS ONE

Journal Requirements:

2. We note that Figure(s) 1 , in your submission contain [map/satellite] images which may be copyrighted. All PLOS content is published under the Creative Commons Attribution License (CC BY 4.0), which means that the manuscript, images, and Supporting Information files will be freely available online, and any third party is permitted to access, download, copy, distribute, and use these materials in any way, even commercially, with proper attribution. For these reasons, we cannot publish previously copyrighted maps or satellite images created using proprietary data, such as Google software (Google Maps, Street View, and Earth). For more information, see our copyright guidelines: http://journals.plos.org/plosone/s/licenses-and-copyright.

1. You may seek permission from the original copyright holder of Figure(s) [#] to publish the content specifically under the CC BY 4.0 license.  

"Funding: The authors thank their home institutions for providing support"

"The authors thank their home institutions for providing support"

Reviewers' comments:

Reviewer's Responses to Questions

**Comments to the Author**

1. Is the manuscript technically sound, and do the data support the conclusions?

Reviewer #1: Yes

Reviewer #2: Yes

2. Has the statistical analysis been performed appropriately and rigorously? 

Reviewer #1: Yes

Reviewer #2: Yes

3. Have the authors made all data underlying the findings in their manuscript fully available?

Reviewer #1: Yes

Reviewer #2: Yes

4. Is the manuscript presented in an intelligible fashion and written in standard English?

Reviewer #1: Yes

Reviewer #2: Yes

5. Review Comments to the Author

Reviewer #1: The authors of this manuscript have produced a useful tool for generating provenanced range maps so that they can be used to inform conservation assessments. However, despite the apparent utility of the tool to batch process large numbers of species, the authors repeatedly mention the need for closer inspection of the data in order to filter out incorrect records. The title and the abstract both emphasise the batch processing aspects, but much of the main text refers to the need for careful inspection of the data, so it is not clear how much of the process is automated and how much additional time is needed to manually correct data. It would be good of the authors could provide some estimates of how much time it takes to collate the data sets, how long it takes the protocol to do the batch processing, and how much time has to be spent manually checking and correcting data.

The authors do admit to the concerns about the quality and reliability of the eBird data (lines 102-107) and explain that the protocol does allow users to check each data point to make decisions about its veracity. However, I feel that the authors have under-stated this issue and it may be a significant issue in other parts of the world with many amateur observers who use eBird as their personal listing software and who are not careful about recording the exact location of each observation and instead use it to record their observations over a much larger area than actually indicated. A stronger caveat on the reliability and quality of the data is required. It is not clear from the analysis to what extent the authors of the paper have inspected all the outlier records to check on their validity, as this is not stated I would assume not. Hence the results showing significant differences between the published ranges and the Areas of Habitat may be due to the inclusion of incorrect records and this needs to be clearly indicated. However, that said, it is clear from the results obtained that there are a number of cases where these differences warrant a re-evaluation of the published ranges. But it would help if the authors were a bit more up-front about the issues of using eBird (and other citizen science data – e.g. the iNaturalist data uploaded to GBIF may even be worse because of issues concerning how records are assigned ‘research grade’ status).

Applying the methodology to birds in other habitats clearly has its limitations as indicated by the authors (lines 384-416). It is even more limited for other taxonomic groups given the patchy nature of records held by other citizen science initiatives. The taxonomic issues (lines 429-444) will pose problems for all taxonomic groups and only the adoption of a common taxonomy would resolve this (see Garnett et. al. 2020 - https://journals.plos.org/plosbiology/article?id=10.1371/journal.pbio.3000736). Despite the limitations it is still a useful proof of concept and hopefully it will be easy to modify the code to allow users to add additional data sources in the future (as indicated in lines 374-382).

A minor point – on line 73, “Extent of Occurrence” is used, however, the IUCN Red List Categories and Criteria (https://portals.iucn.org/library/node/10315) and the Guidelines for Using the IUCN Red List Categories and Criteria (https://nc.iucnredlist.org/redlist/content/attachment_files/RedListGuidelines.pdf) both use “extent of occurrence”.

The authors are also to be congratulated on creating a tool that produces interactive HTML maps allowing users to zoom in on details and the opportunity to check individual data points and their metadata, etc. This kind of functionality helps to increase user confidence and certainly adds to the transparency of the process. A clear advantage of the protocol is its transparency and that it is explicit about what we know and what we don’t know.

I support the publication of this manuscript with some minor corrections to expand on the underlying caveats concerning the use of the eBird data, how automated the process really is, and how easy it can be extended to other taxonomic groups and data sources.

Reviewer #2: # Summary to author

As someone who has recently been working a lot on refining Red List/BirdLife range maps to Area of Habitat, I think anything that makes the process easier and more repeatable is a good thing. Thus, I like the underlying tools of this paper, particularly the transparency of the map-making process and its ability to be refined. The paper is generally well written and the examples given are informative. I really like the integration of eBird data - this is a great idea, and a huge advance.

Where I think the paper currently falls down is the justification for certain limitations of the study. In particular, I don’t follow the restriction to forest-dwelling species. I understand that forest cover data is widely available globally, at a high resolution, but why is that strictly necessary? Why didn’t the authors include all habitat types using, for example, the 2020 Jung et al. IUCN habitat types map? This would of course only give habitat presence/absence in a cell rather that any continuous measure, but then it could be applied at a much larger scale, to all species. I am not requesting that the authors completely re-do this analysis, but rather that they better justify in the text why the focus is on forest species, and how this limitation can/will be overcome in the future. There is some discussion of this point (e.g. lines 89-91), but I think it is insufficient at present.

My other comments are relatively minor, please see below. I think this is a very useful contribution to the field!

# Methods

Lines 114-118: I’m not entirely following this. You identified open habitat species using BirdLife habitat descriptions? And had made the decision a priori to exclude them because they occupy low forest cover habitat? Or you identified that you should exclude these species based on the discovery that they occupy low forest cover habitat?

Lines 129-131: this doesn’t make it clear what you ultimately did with these 99 species

Lines 138-139: can you add a citation for eBird’s best practices?

Line 144: I think more upfront explanation on what an alpha hull actually is (and why you used it) would be useful. Probably just move lines 224-227 up to here.

Line 167: be explicit in what these four values are (your min & max based on GLOBE, plus the Red List min and max?)

Lines 172-173: why 75%? I presume this is an arbitrary value that can be changed by the user?

Lines 178-179: this is crucial, so I think it should be more detailed. Or maybe have a flowchart? If I am following this correctly, the process is: (1) the alpha hull is created based on the range and occurrence points? (N.B. I don’t know what “median inter-presence distance” as the alpha actually means). (2) Min/max elevation and forest cover are identified based on distribution of occurrence points. (3) Alpha is refined to forest cover and elevation that falls within the range identified by (2)? (4) Within this AOH, you identify cells that are potentially occupied/unoccupied based solely on their proximity to presence/absence points, not anything to do with the environmental characteristics of that cell?

Line 184: what does “previously calculated” mean? By whom? You or the Red List? Or both, as in step (2) above?

Lines 185-186: I’m not sure what this means. Species would not be included on the checklist in cells where the habitat is thought to be unsuitable? Is the point that presences outside of range polygon can be included, while absences cannot (because they wouldn’t appear on the checklist)?

# Results

Line 230: many of the circle points also show a wide scatter

Lines 239-241: perhaps you should be more careful in the way the “AOH 12% larger than published ranges” finding is presented, since it includes the potentially unsuitable areas as well. It seems to me that the “potentially suitable AOH” is more accurate, since it is more strongly informed by actual observations. Probably (?) a lot of the “potentially unsuitable AOH” is indeed unsuitable but for reasons we can’t pick up through the environmental layers we currently have, only through repeatedly not observing the species there.

Lines 304-315: very interesting examples. Really exemplifies the utility of this approach.

# Discussion

Lines 3421-343: as above, I think more caution should be added to this point.

Lines 391-393: this starts to get at my concern discussed above re focus on forest species, but I’m not quite convinced. Surely you could limit by habitat, it would just be binary (habitat is present/absent in that cell) rather than a continuous measure like forest cover? But that will surely be true for a long time to come, because any continuous measure of non-forest “quality” is hard to measure remotely? Basically, I am concerned that we will be neglecting many non-forest species globally and for a long time if we ignore them until we have data as good as we have for forest species.

Lines 434-438: I would suggest adding some kind of text ref here before the numeric “[1]”, so that it is easier to read

Lines 457-458: could you add a very brief summary of the supplementary findings here for the lazy/busy reader?

# Figures and tables

As a general point, the figures are a bit low res at present. Please ensure they are provided at a higher resolution for publication.

Figure 1: in the legend it refers to blue lines, but I can’t really make them out in the figure.

Figure 3: please explain the solid black dots in more detail, in both the figure legend and its caption.

6. PLOS authors have the option to publish the peer review history of their article (what does this mean?). If published, this will include your full peer review and any attached files.

Reviewer #1: No

Reviewer #2: No

---

## [Author Response · Author response to Decision Letter 0]

14 Oct 2021

Journal Requirements:

2. We note that Figure(s) 1 , in your submission contain [map/satellite] images which may be copyrighted. All PLOS content is published under the Creative Commons Attribution License (CC BY 4.0), which means that the manuscript, images, and Supporting Information files will be freely available online, and any third party is permitted to access, download, copy, distribute, and use these materials in any way, even commercially, with proper attribution. For these reasons, we cannot publish previously copyrighted maps or satellite images created using proprietary data, such as Google software (Google Maps, Street View, and Earth). For more information, see our copyright guidelines: http://journals.plos.org/plosone/s/licenses-and-copyright.

1. You may seek permission from the original copyright holder of Figure(s) [#] to publish the content specifically under the CC BY 4.0 license. 

The original figure attributions were incorrect and the basemaps were derived from USGS data and the captions now reflect the correction. The only exception is Figure 2 which uses an ESRI basemap for which ESRI provides permission for reproduction for research purposes: https://www.esri.com/en-us/legal/copyright-trademarks

"Funding: The authors thank their home institutions for providing support"

"The authors thank their home institutions for providing support"

We have now removed all funding references from the Acknowledgements section

Reviewer #1: The authors of this manuscript have produced a useful tool for generating provenanced range maps so that they can be used to inform conservation assessments. However, despite the apparent utility of the tool to batch process large numbers of species, the authors repeatedly mention the need for closer inspection of the data in order to filter out incorrect records. The title and the abstract both emphasise the batch processing aspects, but much of the main text refers to the need for careful inspection of the data, so it is not clear how much of the process is automated and how much additional time is needed to manually correct data. It would be good of the authors could provide some estimates of how much time it takes to collate the data sets, how long it takes the protocol to do the batch processing, and how much time has to be spent manually checking and correcting data.

The authors do admit to the concerns about the quality and reliability of the eBird data (lines 102-107) and explain that the protocol does allow users to check each data point to make decisions about its veracity. However, I feel that the authors have under-stated this issue and it may be a significant issue in other parts of the world with many amateur observers who use eBird as their personal listing software and who are not careful about recording the exact location of each observation and instead use it to record their observations over a much larger area than actually indicated. A stronger caveat on the reliability and quality of the data is required. It is not clear from the analysis to what extent the authors of the paper have inspected all the outlier records to check on their validity, as this is not stated I would assume not. Hence the results showing significant differences between the published ranges and the Areas of Habitat may be due to the inclusion of incorrect records and this needs to be clearly indicated. However, that said, it is clear from the results obtained that there are a number of cases where these differences warrant a re-evaluation of the published ranges. But it would help if the authors were a bit more up-front about the issues of using eBird (and other citizen science data – e.g. the iNaturalist data uploaded to GBIF may even be worse because of issues concerning how records are assigned ‘research grade’ status).

The filtering methodology described in the paper is the result of spending a significant amount of time reviewing and investigating data for potential outliers. Together with eBird’s existing review process, the data are of higher quality than the reviewer implies here. As a result, the methods presented allow for batch processing. The issue of individual review is dependent on the ultimate use of the maps. Individual review may not be needed when aggregating result across many species, but applications focused on a single or few species will want a detailed review process, which our transparent methodology allows for. We have now included text that says so. 

The science behind iNaturalist was conceived in the lab that produced this current paper and we remain keenly interested in its development. In addition, we are currently examining eBird, iNaturalist, and GBIF data for projects that naturally emerge from this effort. Our experiences are that mapping species distributions in the ways we do here is often the best way to identify outliers and check their plausibility. As we stress continually, transparency — the ability to check who claimed which species, where, and when is vital to such efforts. 

Applying the methodology to birds in other habitats clearly has its limitations as indicated by the authors (lines 384-416). It is even more limited for other taxonomic groups given the patchy nature of records held by other citizen science initiatives. The taxonomic issues (lines 429-444) will pose problems for all taxonomic groups and only the adoption of a common taxonomy would resolve this (see Garnett et. al. 2020 - https://journals.plos.org/plosbiology/article?id=10.1371/journal.pbio.3000736). Despite the limitations it is still a useful proof of concept and hopefully it will be easy to modify the code to allow users to add additional data sources in the future (as indicated in lines 374-382).

We agree! Another motivation of this effort is its shining light on distributions that in turn inform taxonomic issues. In doing so, sometimes records show geographically distinct populations, but in others, continuous distributions for which more detailed studies might show clinal variation. 

A minor point – on line 73, “Extent of Occurrence” is used, however, the IUCN Red List Categories and Criteria (https://portals.iucn.org/library/node/10315) and the Guidelines for Using the IUCN Red List Categories and Criteria (https://nc.iucnredlist.org/redlist/content/attachment_files/RedListGuidelines.pdf) both use “extent of occurrence”.

Changed to lower case

The authors are also to be congratulated on creating a tool that produces interactive HTML maps allowing users to zoom in on details and the opportunity to check individual data points and their metadata, etc. This kind of functionality helps to increase user confidence and certainly adds to the transparency of the process. A clear advantage of the protocol is its transparency and that it is explicit about what we know and what we don’t know.

We appreciate the kind words and are very happy that the reviewer acknowledges the value of transparency!

I support the publication of this manuscript with some minor corrections to expand on the underlying caveats concerning the use of the eBird data, how automated the process really is, and how easy it can be extended to other taxonomic groups and data sources.\\

We have added the relevant caveats and explanations

 

Reviewer #2: # Summary to author

As someone who has recently been working a lot on refining Red List/BirdLife range maps to Area of Habitat, I think anything that makes the process easier and more repeatable is a good thing. Thus, I like the underlying tools of this paper, particularly the transparency of the map-making process and its ability to be refined. The paper is generally well written and the examples given are informative. I really like the integration of eBird data - this is a great idea, and a huge advance.

Where I think the paper currently falls down is the justification for certain limitations of the study. In particular, I don’t follow the restriction to forest-dwelling species. I understand that forest cover data is widely available globally, at a high resolution, but why is that strictly necessary? Why didn’t the authors include all habitat types using, for example, the 2020 Jung et al. IUCN habitat types map? This would of course only give habitat presence/absence in a cell rather that any continuous measure, but then it could be applied at a much larger scale, to all species. I am not requesting that the authors completely re-do this analysis, but rather that they better justify in the text why the focus is on forest species, and how this limitation can/will be overcome in the future. There is some discussion of this point (e.g. lines 89-91), but I think it is insufficient at present.

We are exploring alternatives in current work to improve upon exactly this point. The idea of matching each observation to the mapped habitat at that location is indeed seductive and one of the author’s has done so in a paper in press at Conservation Biology (https://www.biorxiv.org/content/10.1101/2021.06.08.447053v1). Nonetheless, our experiences are (1) the habitat classification categories are too coarse, (2) when mapped, grasslands and other open area habitats are still subject to errors in accuracy, and (3) that even when one gets past these issues, the connection to species preferences is difficult. Jung et al. include one of this paper’s authors and we are familiar with it. 

(1) Many of the habitat classification/global land cover data, such as the Jung et al or ESA CCI maps, divide the land into a relatively few number of categories. Instead, using a continuous scale of forest cover allows us to create maps of species ranges at a finer resolution which we believe is appropriate for what we consider the first version of a proof of concept.

(2) Accurate remote sensing and mapping of grasslands and their threats is difficult. Even by Jung et al’s own validation, their accuracy for many grassland and shrubland level 2 habitats is only ~50%. We do not intend for this to be a limit to our methodologies, but the resolution of such a challenge is beyond the scope of this paper.

(3) In current work, we are examining the habitat choices of ~100 endemic species that occur across a wide range of habitats. We ask which species occurrences are in habitats more often than one would expect by chance, given the habitat’s extent. One quickly sees that while there are some clear preferences, such habitats sometimes do not account for more than two-thirds of the species occurrences. So does one only include habitats in which the species occurs more often than by chance, or all habitats to include (say) 95% of the occurrences. The former has large errors of omission, the latter large errors of commission. Similarly, were we to use Jung et al.’s classification, would we use all habitats or only those top preferences? 

Given these complications, we find the rigorously quantified measurement of forest cover provides a simple way forward. It’s one for which we can readily explain the consequences of changing the thresholds we employ. Yes, we do not cover all the species — which is why we are exploring extensions to our methods in current work. At the moment, we have added some text in the Limitations mentioning errors of omission and commission in a habitat class matching approach.

My other comments are relatively minor, please see below. I think this is a very useful contribution to the field!

# Methods

Lines 114-118: I’m not entirely following this. You identified open habitat species using BirdLife habitat descriptions? And had made the decision a priori to exclude them because they occupy low forest cover habitat? Or you identified that you should exclude these species based on the discovery that they occupy low forest cover habitat?

Because of the batch nature of our methods, we ran all the species based on range size and listed forest habitat cover statistics. When we did so, it was immediately apparent that some species were outliers — most of them known from our personal knowledge. Miners, cinclodes, and helmetcrest hummingbirds were obvious examples. With that insight, we then went to the BirdLife classifications and used the first-mentioned major habitat to decide whether the species was predominantly a grassland or shrubland species. 

Lines 129-131: this doesn’t make it clear what you ultimately did with these 99 species

Now clarified

Lines 138-139: can you add a citation for eBird’s best practices?

Now included

Line 144: I think more upfront explanation on what an alpha hull actually is (and why you used it) would be useful. Probably just move lines 224-227 up to here.

Moved and expounded

Line 167: be explicit in what these four values are (your min & max based on GLOBE, plus the Red List min and max?)

Done

Lines 172-173: why 75%? I presume this is an arbitrary value that can be changed by the user?

That is correct

Lines 178-179: this is crucial, so I think it should be more detailed. Or maybe have a flowchart? If I am following this correctly, the process is: (1) the alpha hull is created based on the range and occurrence points? (N.B. I don’t know what “median inter-presence distance” as the alpha actually means). (2) Min/max elevation and forest cover are identified based on distribution of occurrence points. (3) Alpha is refined to forest cover and elevation that falls within the range identified by (2)? (4) Within this AOH, you identify cells that are potentially occupied/unoccupied based solely on their proximity to presence/absence points, not anything to do with the environmental characteristics of that cell?

This is all correct. Figure 1 is intended to serve as a flow chart analog with a visual example and we believe a separate flowchart would be redundant. That being said, we have expounded on the alpha hull description for a clearer understanding for your N.B.

Line 184: what does “previously calculated” mean? By whom? You or the Red List? Or both, as in step (2) above?

Clarified in text

Lines 185-186: I’m not sure what this means. Species would not be included on the checklist in cells where the habitat is thought to be unsuitable? Is the point that presences outside of range polygon can be included, while absences cannot (because they wouldn’t appear on the checklist)?

That is not the correct interpretation and we’ve now tried to clarify in the text. We don’t count checklists in unsuitable habitat for absences, but we do for presences since we are more conservative about what constitutes a true absence, but have higher confidence in what is considered a true presence.

# Results

Line 230: many of the circle points also show a wide scatter

We wrote “a wider scatter” — which is correct, but yes, all the points have a wide scatter. 

Lines 239-241: perhaps you should be more careful in the way the “AOH 12% larger than published ranges” finding is presented, since it includes the potentially unsuitable areas as well. It seems to me that the “potentially suitable AOH” is more accurate, since it is more strongly informed by actual observations. Probably (?) a lot of the “potentially unsuitable AOH” is indeed unsuitable but for reasons we can’t pick up through the environmental layers we currently have, only through repeatedly not observing the species there.

The division in the two types of AOH are not based on “suitability” but rather potential occupancy. AOH by definition is all the area that appears to be suitable elevation and landcover and is originally defined in Brooks et al. 2019. For this reason we have chosen to keep the text unchanged.

Lines 304-315: very interesting examples. Really exemplifies the utility of this approach.

Thanks!

# Discussion

Lines 3421-343: as above, I think more caution should be added to this point.

See above response

Lines 391-393: this starts to get at my concern discussed above re focus on forest species, but I’m not quite convinced. Surely you could limit by habitat, it would just be binary (habitat is present/absent in that cell) rather than a continuous measure like forest cover? But that will surely be true for a long time to come, because any continuous measure of non-forest “quality” is hard to measure remotely? Basically, I am concerned that we will be neglecting many non-forest species globally and for a long time if we ignore them until we have data as good as we have for forest species.

Exactly so! There are 15 open habitat species that are threatened, of which the white-bellied cinclodes is critically endangered. Our using that as an example is not just because Pimm has a photograph of it. Based on habitat maps, this species should have a large amount of habitat. Clearly most of that is unsuitable, so existing habitat maps are a poor guide. (The species also occurs more widely than previously known, showing the importance of crowd-sourced data.). Mapping grassland species from remote sensing requires considerable care, as Pimm’s work on the Cape Sable seaside sparrow shows. Such efforts are likely to be one-off, specialized maps, so the best we can do is to acknowledge existing limitations. 

Lines 434-438: I would suggest adding some kind of text ref here before the numeric “[1]”, so that it is easier to read

We’ve now added the subject

Lines 457-458: could you add a very brief summary of the supplementary findings here for the lazy/busy reader?

We have added a couple of sentences to this point.

# Figures and tables

As a general point, the figures are a bit low res at present. Please ensure they are provided at a higher resolution for publication.

Figure 1: in the legend it refers to blue lines, but I can’t really make them out in the figure.

Made clearer

Figure 3: please explain the solid black dots in more detail, in both the figure legend and its caption.

Explained

---

## [Editor Report · Decision Letter 1]

18 Oct 2021

Batch-produced, GIS-informed range maps for birds based on provenanced, crowd-sourced data inform conservation assessments

PONE-D-21-23994R1

Dear Stuart,

We’re pleased to inform you that your manuscript has been judged scientifically suitable for publication and will be formally accepted for publication once it meets all outstanding technical requirements.

Kind regards,

Tim A. Mousseau

Academic Editor

PLOS ONE

---

## [Editor Report · Acceptance letter]

3 Nov 2021

PONE-D-21-23994R1 

Batch-produced, GIS-informed range maps for birds based on provenanced, crowd-sourced data inform conservation assessments 

Dear Dr. Pimm:

I'm pleased to inform you that your manuscript has been deemed suitable for publication in PLOS ONE. Congratulations! Your manuscript is now with our production department. 

Kind regards, 

on behalf of

Dr. Tim A. Mousseau 

Academic Editor

PLOS ONE